# Effects of Non-Thermal Plasma on Yeast *Saccharomyces cerevisiae*

**DOI:** 10.3390/ijms22052247

**Published:** 2021-02-24

**Authors:** Peter Polčic, Zdenko Machala

**Affiliations:** 1Department of Biochemistry, Faculty of Natural Sciences, Comenius University in Bratislava, Mlynská dolina CH1, Ilkovičova 6, 84215 Bratislava, Slovakia; 2Division of Environmental Physics, Faculty of Mathematics, Physics, and Informatics, Comenius University in Bratislava, Mlynská dolina F2, 84248 Bratislava, Slovakia; machala@fmph.uniba.sk

**Keywords:** cold plasma, yeast, oxidative stress

## Abstract

Cold plasmas generated by various electrical discharges can affect cell physiology or induce cell damage that may often result in the loss of viability. Many cold plasma-based technologies have emerged in recent years that are aimed at manipulating the cells within various environments or tissues. These include inactivation of microorganisms for the purpose of sterilization, food processing, induction of seeds germination, but also the treatment of cells in the therapy. Mechanisms that underlie the plasma-cell interactions are, however, still poorly understood. Dissection of cellular pathways or structures affected by plasma using simple eukaryotic models is therefore desirable. Yeast *Saccharomyces cerevisiae* is a traditional model organism with unprecedented impact on our knowledge of processes in eukaryotic cells. As such, it had been also employed in studies of plasma-cell interactions. This review focuses on the effects of cold plasma on yeast cells.

## 1. Introduction

Non-thermal (cold) plasmas, also referred to as non-equilibrium plasmas, can be generated by various types of electrical discharges, such as coronas, sparks, dielectric barrier or radio-frequency discharges, at low, as well as at atmospheric pressure. They are ionized gases, in which the electrons acquire high energy, but the gas particles retain a relatively low, often even ambient, temperature. Exposure to non-thermal (cold) plasmas is known to affect living cells in a way that depends on plasma characteristics and dose, as well as on the cell type. Since the treatment of cells by cold plasma may produce desirable effects, a number of practical applications have arisen from investigation of plasma-cells interactions. These include inactivation of microorganisms for the purpose of disinfection or sterilization [1,2,3,4,5], food treatment to prevent spoilage [6] or induction of germination in seeds and enhanced plant growth [7]. Treatment by plasma has been proven to support wound healing [8] and the plasma devices for such treatment have already been approved in several countries [9]. The techniques are being developed, by which specific cells, e.g., cancer cells, can be selectively killed. These applications could, thus, potentially be useful for the treatment of various pathologies, such as cancer [1,2,3,4,10,11]. The mechanisms involved in cell killing by cold plasmas are, however, still poorly understood and dissection of cellular pathways as well as identification of cell components affected by plasma using simple eukaryotic models is needed.

Yeast *Saccharomyces cerevisiae* has been a traditional eukaryotic model organism with an unprecedented impact on biochemistry and cell biology [12]. It has been used to characterize countless fundamental processes in eukaryotic cells, including gene expression, the cell cycle progression and membrane trafficking, to name a few. Although yeast may not have some of the pathways present in specialized types of animal cells, they often contain individual components of these pathways conserved and may, thus, be used to study specific subroutines of these pathways, as is in a case of many human diseases, human cell death pathway (apoptosis) [13], or even in the biology of neurons [14] or viruses [15]. Moreover, mammalian (e.g., human) components that are completely absent in yeast can often be expressed in yeast in order to study them in a simple ‘humanized’ yeast model system [16,17]. With a powerful battery of genetic and biochemical methods that cannot be rivalled by any other eukaryotic system, yeast are being also recognized as a perspective model for studying the effects of plasma on cells. This review focuses on our current knowledge of interactions of cold plasma with the cells of this model organism.

## 2. Plasma Treatment of Yeast Cells

Cold plasma has been well characterized as extremely toxic for bacteria, which led to the development of different cold-plasma based technologies for decontamination of various materials [5]. Sensitivity of eukaryotic cells, including yeast, to plasma treatment is generally lower as compared with bacteria, but significant fraction of eukaryotic cells can be killed by cold plasma in a dose-dependent manner (with plasma “dose” usually expressed as the plasma treatment time) [18,19,20]. The effectivity of killing and likely also molecular mechanisms involved in cell killing also depend on the mode of the action of plasma on cells. While many different settings are used to experimentally treat cells by cold plasma, two basic modes of treatment differ significantly. These are the direct treatment of cells, in which cells are exposed to the plasma discharge, or indirect, in which cells are incubated in a liquid medium that was treated by plasma, e.g., plasma-activated water. Somewhere in between these two extremes are the settings (plasma jets), in which a carrier gas moves through the discharge and carries the generated plasma to the treated material (e.g., cells). Examples of typical experimental setups used for treating of cells with cold plasma are shown in Figure 1.

Chemical composition of plasma is rather complex and may differ in different experimental settings, depending on gases in the plasma and surrounding environment. In plasma generated in atmospheric air, a broad array of reactive oxygen species (ROS) and reactive nitrogen species (RNS) can be detected. These include radicals such as hydroxide (^•^OH), oxygen (^•^O), nitrogen (^•^N), nitric oxide (^•^NO), hydroxyperoxyl (HO_2_); ions, e.g., superoxide (O_2_^•−^); and other oxidants, e.g., ozone (O_3_) or singlet oxygen (^1^O_2_) [21,22]. Both in air and in the water treated by plasma, these particles react with each others and with water, so that other particles, such as hydrogen peroxide (H_2_O_2_), nitrite (NO_2_^−^) and nitrate (NO_3_^−^) are generated and the treated water typically gets acidified (pH~3.2, unless buffered) [23]. Short-lived but reactive peroxynitrite (ONOO^−^) has also been detected [23,24].

As one would expect, significantly more yeast cells are killed by direct plasma treatment than when cells are incubated in plasma-activated water [25]. These results indicate that in direct treatment regime, a significant portion of cells dies immediately due to the exposure to the factors present in the electric discharge (e.g., electric field, short-living reactive particles, UV irradiation). Additionally, a portion of cells that survive the discharge dies due to the presence of reactive particles generated in the discharge and surviving in the plasma-activated water for significant periods of time after the discharge. From a comparison of survival rates of directly and indirectly treated cells, it appears that these two components of cell killing effects act together, likely in synergy.

When survival of yeast cells after treatment with He/O_2_ plasma jet was assessed in different surrounding media, the same parameters of plasma were found to induce different changes [26]. The lowest survival rates were observed when cells were treated in water, with the effect being less severe in phosphate buffered saline (PBS) and mildest in YPD (yeast extract, peptone, dextrose), where survival rates were significantly higher. Cell-protective effect of PBS is probably caused by its buffering capacity and salt strength rather than osmotic stabilization, as lowering the salt concentration leads to a lowered protective effect that cannot be reproduced by using sorbitol. Acidification of media appears to play a role, but it must be the combination of pH and the presence of reactive particles that affects the survival, since yeast normally tolerate the corresponding acidic pH [27]. The weak effect of plasma on cells in YPD can, in addition to its buffering capacity, likely be explained by sequestering reactive particles by the components of the medium. For example, if one only considers the reactions of reactive particles from the plasma with proteins, there is roughly 100× more mass of peptides in YPD (2% peptone) than is proteins in treated cells (at 10^7^ cells/mL).

## 3. Plasma-Induced Cell Death in Yeast

Analyses of plasma-treated yeast cells revealed numerous changes. Visualization of treated cells by electron scanning microscopy uncovered changes in cell morphology and in the cell wall. When yeast cells on a filter were treated with He/O_2_ plasma jet, changes resembling peeling have been observed in the cell wall [18]. Different changes were observed when yeast were treated in a suspension. Under these conditions, a portion of treated cells acquired shrunken and crashed appearance, with both the severity and the incidence of changes depending on the surrounding medium [26]. The most severe and frequent changes were observed when cells were treated in water, less severe in saline solution and mildest in YPD, which correlates with the survival rate. Changes in appearance of cells were also observed by transmission electron microscopy.

Since it has been well-established that properly dosed plasma treatment can induce programmed cell death by apoptosis in mammalian cells [4,10,28,29,30], plasma-treated yeast cells were also investigated for the presence of markers of programmed cell death. Although yeast do not have the pathway that directly corresponds to the mammalian apoptotic pathway, it has been generally accepted that yeast cells can undergo regulated cell death and some of the components of yeast cell death pathways are homologous to their mammalian counterparts with well-established roles in apoptosis [31,32,33,34,35]. Interestingly, several changes that are considered as typical hallmarks of regulated cell death have been observed in yeast cells treated with He/O_2_ plasma jet [36] or directly by a corona discharge [37]. These changes involve generation of a population of cells that are stained with Annexin-V, due to the exposure of phosphatidylserine at the cell surface, while they are not stained with propidium iodide, indicating the intact plasma membrane [36,37]; chromatin condensation that can be visualized by staining with DAPI (4′,6-diamidino-2-phenylindole); decrease in mitochondrial transmembrane potential revealed by staining with TMRM (tetramethylrhodamine methyl ester) and cell cycle arrest in G1 phase [36]. These observations may indicate that, at least partially, the dying observed in yeast after the plasma treatment may be attributed to regulated cell death that is triggered as a reaction to, either cell damage or to the presence of reactive particles originating from plasma itself. However, when the participation of known components of yeast cell death pathway on the plasma-induced dying was analyzed, it was found that mutant yeast strains that lack these components retain unchanged death rate as compared with the wild type strains [25]. The tested yeast cell death components involved the yeast metacaspase Yca1p [38], yeast homologue of mammalian apoptosis-inducing factor Aif1p [39], and endonuclease G Nuc1p [40]. Contrary to the detection of apoptotic markers, these results indicate that participation of Yca1p, Aif1p and Nuc1p is not required for plasma-induced cell death as it would be expected in the case of plasma-induced apoptosis. Dying of cells is thus likely directly caused by malfunction of vital cellular components due to the damage caused by plasma rather than a cell response to the damage by the activation of cell death program. The presence of apoptotic markers in dying cells may result from activation of some of the downstream subroutines of regulated cell death pathway that may in this case accompany the unregulated death. However, it was not ruled out that regulated cell death pathways that do not rely on function of Yca1p, Aif1p and Nuc1p may exist in yeast and be activated in the cells damaged by plasma. Bacause in apoptosis, or in other forms of programmed cell death, cells are dying due to the execution of intrinsic cell killing program rather than directly due to the damage, to prove that apoptosis participates in cell killing by plasma, it has to be shown that blocking of participating cell death pathway would result in an increased cell survival after treatment with plasma.

## 4. Effect of Plasma on Membranes and Energy Metabolism

Membrane lipids are well known target of oxidative damage in cells. Oxidants, such as free radicals, react with lipids containing one or more carbon-carbon double bond, especially polyunsaturated fatty acids (PUFAs), resulting in lipid peroxidation. This process not only leads to the damage of cell membranes but also generates further toxic reactive species (see [41] for review). One of the major products of lipid peroxidation is malonedialdehyde (MDA), which is also often used as a marker of lipid peroxidation as it can be easily quantified by a reaction with thiobarbituric acid [42]. Production of MDA has been previously described in plasma-treated bacteria *Escherichia coli* [19,43,44].

Yeast *S. cereviseae* (similar to *E. coli*) do not produce polyunsaturated fatty acids with more than one double bond [45,46], but the generation of MDA was described in the plasma-treated yeast cells [36,47], indicating that yeast lipids get peroxidated under these conditions. In the same study, it was also shown that the same treatment leads to the leakage of proteins as well as potassium ions (K^+^) from the treated cells to the surrounding medium, to the loss of the membrane potential at cytoplasmic membrane and to an increase in the intracellular pH. These findings together clearly indicate that the plasma treatment disrupts the integrity of yeast cytoplasmic membrane.

Although it has not been specifically shown that intracellular membranes of yeast are damaged by plasma, it is highly probable. An increase in intracellular Ca^2+^ observed in plasma-treated cells [47] can likely be attributed to the disruption of the barrier function of cellular membranes. Staining of yeast cells with TMRM or JC-1 further indicated that, in plasma-treated yeast cells, mitochondria may not be able to maintain transmembrane potential (Δ*ψ*) [36,47], resulting in a dramatic decrease in cellular ATP level [47]. This may be, indeed, caused by the damage to the inner mitochondrial membrane due to lipid peroxidation. On the other hand, numerous causes other than membrane damage could explain the loss of membrane potential, including the changes in metabolism of the cell due to oxidative damage of many cellular enzymes or, as it has also been suggested [47], by opening of the mitochondrial permeability transition pore (PTP). As the latter is a high-conductance nonspecific channel in mitochondrial membranes that opens in response to Ca^2+^ [48], its opening and participation on the dissipation of mitochondrial Δ*ψ* is consistent with the increase in intracellular Ca^2+^. Here, however, it should be mentioned that PTP in yeast mitochondria differs from that known in mammalian cells in several aspects. One of these is that due to the absence of Ca^2+^ uptake transporter in yeast mitochondria, yeast PTP can only be opened in vitro (in isolated mitochondria) in the presence of a Ca^2+^-ionophore [49]. Although the role of PTP in cell death in yeast has been suggested, mostly based on the analogy with mammalian cells [50], there is no evidence so far of PTP opening in yeast in vivo nor its participation in any physiological process in yeast has been proven.

It may be interesting to note here that most of the data on ion homeostasis, pH and transmembrane potential are based on the measurements that rely on fluorescent probes and their accumulation in relevant cell compartments. Some of these data, thus, may be significantly biased by the disruption of the integrity of membranes that clearly happens upon the plasma treatment.

## 5. Plasma Damage to DNA

Cellular DNA represents another well-established target of plasma. Chemical changes induced by plasma treatment of isolated DNA have been intensively studied and involve the formation of DNA strand breaks, dimerization and modification of bases [51]. In eukaryotic cells, the formation of single-strand and double-strand breaks [51,52], as well as creation of oxidized bases such as 8-oxodeoxyguanosine [51,53], has been described. These changes are subject to DNA repair mechanisms but may result in mutations, or in cell death if the damage exceeds the capacity of the DNA repair system. Based on the ability of plasma to induce mutations, a technique called atmospheric and room temperature plasmas (ARTP) mutagenesis has been developed [54]. The ARTP mutagenesis, which can generate higher mutation rate than traditional random mutagenesis techniques, such as UV irradiation or chemical mutagenesis, was successfully used to generate mutants of several microorganisms, including bacteria, yeasts and algae, mostly for the purpose of generating strains with desired biotechnological properties [55,56].

To study the possible effect of plasma on the chromosomal stability in yeast, a sensitivity of wild-type and *rad51* mutant to the plasma treatment was measured. *RAD51* encodes for a strand exchange protein that is required for effective homologous recombination in the process of DNA damage repair. Yeast strains lacking a functional Rad51p are therefore hypersensitive to agents that induce the formation of double-strand breaks in DNA [57]. When treated with non-thermal atmospheric plasma, Rad51-defficient mutant was found to be significantly more sensitive as compared with wild type, indicating that the reparation of double-strand breaks contributes to the survival after plasma treatment [58]. To confirm that the increased sensitivity of this mutant to plasma really is a result of defective homologous recombination, two other deletion mutants lacking proteins participating in the same pathway have been tested for sensitivity to plasma treatment [58]. Both mutants, lacking either Rad52p, a protein of the same epistatic group as Rad51p, that facilitates the Rad51p binding to DNA [59,60], or Mec1p, a protein kinase required for DNA damage checkpoint (yeast analog of human ATR) [61,62], were also found to be more sensitive to the air plasma treatment. Moreover, when yeast cells expressing Rad52-GFP were treated with plasma, green fluorescent foci were observed in cell nuclei, resembling those induced by phleomycin, an antibiotic known to induce double-strand breaks in DNA. These foci represent the Rad52-GFP sequestered on the damaged DNA. Their formation thus indicates that the plasma-induced damage in DNA is actually recognized by Rad52p. Interestingly, the induction of double-strand breaks that happens in atmospheric air plasma was not observed when plasma was generated in N_2_, indicating that the particles that react with DNA to cause this type of damage may not be present in plasma under these conditions [58].

Another type of DNA damage that was detected in plasma-treated yeast cells is the formation of DNA-protein crosslinks [63]. This is a specific type of DNA damage, in which proteins are covalently bound to DNA. As the resulting lesions can be bulky, this type of DNA damage usually interferes with transcription and replication [64]. In the case of plasma treatment, the nuclear proteins, e.g., histones, are likely attacked by ROS and RNS originating in plasma to form reactive intermediates that then react with DNA [63].

The ability of plasma to induce the formation of DNA-protein crosslinks in cells may be particularly interesting from the perspective of the use of plasma in cancer therapy because induction of DNA-protein crosslinks is particularly toxic for dividing cells and represents a mechanism, by which several anti-cancer chemotherapeuticals work. Although these drugs mostly act by binding the specific enzymes to DNA, some of them, such as *cis*-platin, have been shown to crosslink histones to DNA [65,66].

## 6. Genetic Dissection of Plasma-Induced Changes in Yeast

Yeast *S. cerevisiae* is the first eukaryotic organism, the genome of which was completely sequenced [67]. Mutants with deletion of any non-essential gene are either available from systematic genomic studies, such as Euroscarf collection [68], or can be easily prepared in any genetic background thanks to the high efficiency of homologous recombination in yeast [69,70]. Testing of the sensitivity of such mutants thus appears a reasonable strategy, by which enzymes or cellular components that affect the survival after plasma treatment may be identified. Using this approach, several genes have been found to affect the survival of cells after treatment with plasma.

Since the major cell killing effect of the plasma is generally attributed to ROS and RNS, deletion strains lacking enzymes that inactivate such reactive particles have been tested in the plasma treatment. There are two superoxide dismutases in *S. cerevisiae*. The cytosolic Cu/Zn-superoxide dismutase Sod1p and the mitochondrial Mn-containing Sod2p are components of an inducible system of cellular protection against superoxide, but expression and activities of both are also detectable in exponentially growing cells [71,72,73]. Similarly, the protection from hydrogen peroxide is provided by two catalases, cytosolic Ctt1p [74] and peroxisomal Cta1p [75,76]. While yeast strains lacking either one of the two superoxide dismutases (SODs) were found to be more sensitive both towards the direct plasma treatment and towards the incubation in the plasma-activated water, mutants lacking any of the two catalases did not differ from the wild type control [25]. Higher sensitivity of SOD-deletion mutants suggests that superoxide must play a significant role in killing cells by plasma, and that the superoxide dismutases expressed in cells are able to considerably reduce the concentration of superoxide. These results are perfectly consistent with increased plasma resistance observed in strains overexpressing *SOD1* or *SOD2* [36]. On the other hand, the negligible effect of the deletion of the genes encoding for catalases on survival after the plasma treatment supports the idea that hydrogen peroxide does not play a key role in plasma-induced killing of yeast [25]. However, here, one has to be cautious in interpreting this result in such a way, as the deletion of *CTT1* and *CTA1* generally does not affect the survival of yeast when treated with comparable concentrations of hydrogen peroxide [77].

In another study that employed a similar approach, plasma jet treatment was applied to a set of mutants with individual deletions of genes participating in oxidative stress response and cell cycle regulating pathways. Several of the tested mutants were found to be more sensitive to plasma than wild type [78]. From the oxidative stress pathway, these were deletions of genes *HOG1*, *SSK2*, *SKN7* and *ASK10*. Products of these genes participate at stress responsive Mitogen-activated protein kinase (MAPK) cascade, an evolutionarily conserved pathway that signals the stress to the cell nucleus and governs the transcriptional regulation of many genes involved in stress response [79]. In this cascade, Hog1p represents a key protein kinase (MAPK) that upon its activation enters the cell nucleus and phosphorylates a repressor protein Sko1p to inactivate it. This in turn activates the expression (transcription) of target genes [80,81]. Hog1p is activated by upstream kinases Pbs2p (MAPK kinase, MAPKK) and either of two MAPKK kinases (MAPKKK) Ssk2p or Ssk22p. Signalization of stress to the MAPK cascade is mediated by a pathway referred to as multistep phosphorelay, which is a yeast homologue of prokaryotic two-component pathway [82]. It consists of a membrane sensor kinase Sln1p, phosphate transfer protein Ypd1p and Ssk1p that directly regulates Ssk2p. Increased sensitivity of mutants lacking Hog1p or Ssk2p indicates that signaling by MAPK cascade is involved in the response of cells to plasma. The participation of this MAPK cascade was further corroborated by detection of phosphorylation of Hog1p in plasma-treated wild type cells [26]. Interestingly, while three major MAP kinases exist in yeast, Fus3p, Hog1p, and Slt2p, no phosphorylation of Fus3p and Slt2p was induced under same conditions.

Skn7p is another target of multistep phosphorelay [83]. It is a stress-responsive transcription factor [79,84,85] that regulates the oxidative stress-induced expression of anti-oxidant genes, including cytosolic catalase (*CTT1*), cytosolic superoxide dismutase (*SOD1*) and others [86]. Under the conditions of stress, Skn7p is also known to be regulated by Ask10p [87,88]. Decreased resistance of *skn7* and *ask10* mutants to plasma indicates that, among other stress response pathways, this pathway plays a consequential role in cell survival after the plasma treatment [78].

All four mutants defective in genes required for the regulation of cell cycle that were investigated, *CDC28*, *CLN3*, *SWI4* and *SWI6,* were found to be hypersensitive to plasma treatment [78]. Cdc28p is a catalytic subunit of cyclin-dependent kinase that throughout the progression of cell cycle associates with cyclins, such as Cln3p, that regulate its activity and selectivity [89]. The expression of genes required for S phase relies on transcriptional activators known as SBF (SCB-binding factor) and MBF (MCB-binding factor). These factors are heterodimers of Swi4p/Swi6p and Mbp1p/Swi6p, respectively, and their activity is regulated by binding of Whi5p repressor [90,91]. To proceed from G1 to S phase, Cln3p/Cdc28p complex must phosphorylate the Whi5p, which then releases active SBF and MBF to activate the transcription of genes containing SCB and MCB elements in their promoters [92].

Increased sensitivity of mutants with deleted *CLN3*, *SWI4, SWI6* or diminished expression of *CDC28* (which is an essential gene) suggests that flawless regulation of cell cycle may be important for the ability of cells to deal with stresses induced by plasma. It may likely be attributed to the DNA damage checkpoint failure in mutant yeast strains upon the plasma-induced damage of DNA. In the case of *swi4* and *swi6* mutants, this seems to be in agreement with the fact that Swi6p is a target of regulation by Mec1p (see above). In response to DNA damage, Mec1p phosphorylates the protein kinase Rad53p, which in turn phosphorylates Swi6p [93,94]. A lack of Swi6p phosphorylation due to mutation in its phosphorylation site is known to shorten a delay at the checkpoint in case of DNA damage [94]. The lack of the proper response to DNA damage induced by plasma may thus explain decreased survival of *swi6* and *swi4* mutants. Another possible explanation for increased sensitivity of *swi6* mutant may not involve the DNA damage, but rather reflect the fact that Swi6p also acts as an oxidative stress sensor by directly sensing redox status through a specific cysteine residue [95].

As the diminished expression of *CDC28* is known not to increase the sensitivity of cells to DNA damaging agents [96], failure of the checkpoint does not explain the increased plasma sensitivity of *cdc28* and *cln3* mutants. It is, thus, possible that, in addition to the proper function of G1/S checkpoint, which provides the cell with the chance to repair the damage before proceeding to S phase, proper regulation of some additional, yet to be identified, events in a cell cycle is crucial for the recovery from the plasma-induced stress.

## 7. Plasma Treatment Induces Multiple Stress-Response Pathways

When the response of yeast cells to cold plasma was investigated using the GFP-labelled reporter fusion proteins, activation of multiple stress response pathways has been detected. Yap1p is a leucine zipper transcription factor that is known to translocate from the cytoplasm to the nucleus under conditions of oxidative stress, e.g., after H_2_O_2_ treatment, to activate the transcription of anti-oxidant stress response genes [97]. It directly reacts to the oxidative status within the cell as its localization is governed by creation of disulfide bonds within specific region of the protein that also contains a nuclear localization signal [98,99]. When localization of Yap1-GFP was investigated in plasma-treated cells, a significant portion of green fluorescence was found to relocate from the cytoplasm to the cell nucleus within minutes after the plasma treatment [100]. This observation indicates that plasma treatment induces the oxidative stress in yeast cells that is sensed by Yap1p.

Msn2p and Msn4p are the general stress response factors that are stochastically distributed between the cell cytoplasm and nucleus in normal (non-stress) conditions, but increase their relative distribution to the nucleus in response to various stress conditions, including heat shock, osmotic shock, oxidative stress and low pH [101,102]. In the nucleus, these factors activate the transcription of genes containing the stress response element STRE [101,103]. In plasma-treated cells, fusion proteins containing either of these two proteins and green fluorescent protein, Msn2-GFP and Msn4-GFP, translocate from the cytoplasm to the nucleus [100]. As it can be expected, the expression of two genes encoding for heat shock proteins *HSP30* and *SSA4*, that are involved in this stress response pathway and are the target genes of Msn2p and Msn4p (and contain STRE) [104,105], is induced in plasma-treated cells [100]. This clearly indicates that the signalization by this pathway in fact does affect the expression of genes under conditions of plasma treatment.

Chaperones that are involved either in protein folding or in protein maintenance under various stress conditions and have been shown to respond to plasma treatment involve Hsp104p [100], Tsa1p and Ssa1p [106]. Hsp104p is a heat shock protein known to facilitate the disassembly of protein aggregates that are formed under stress conditions [107]. Aggregates of Hsp104-GFP, similar to those formed in cells after a heat shock, can be observed in the cytoplasm of yeast cells treated with plasma [100]. Moreover, the production of centrifugable insoluble protein aggregates, as well as protein ubiquitination, both in a fashion similar to that observed after heat-shock, were detected in the same study. Tsa1p is a cytosolic thioredoxin peroxidase that participates at the protection of cells against oxidative stress [108,109]. Ssa1p is a heat shock protein, a cytosolic member of HSP70 family, that assists in protein folding, prevents protein aggregation or disassembles protein aggregates [110,111]. Tsa1-GFP and Ssa1-GFP fusions were shown to form aggregates in the cytosol in response to plasma treatment [106]. Unlike with Hsp104p, similar change is not induced by heat shock but can be induced by treatment of cells with H_2_O_2_.

Accumulation of unfolded or unproperly folded proteins in endoplasmic reticulum (ER) is referred to as stress of endoplasmic reticulum (ER stress). It can be experimentally induced, e.g., by heat shock or by treatment of cells with reducing agents, such as dithiotreitol (DTT) or β-mercaptoethanol. From the pathways known to respond to ER stress, pathway that relies on Ire1 is the only one described in yeast [112,113]. Ire1p is a transmembrane protein in the membrane of ER that acts as a sensor of accumulation of unfolded proteins in the lumen of ER [114]. Signalization by Ire1p is triggered by the dimerization that leads to the autophosphorylation and activation of its ribonuclease cytosolic domain. Processing of a specific pre-mRNA (HAC1U) by active Ire1p ribonuclease then enables the synthesis of Hac1p transcription factor that affects the expression of relevant genes [115]. The dimerization of IreIp is believed to result from the lack of the binding of chaperone BiP to the ER luminal domain of Ire1p, which occurs when all of the BiP is engaged in binding to unfolded proteins in ER lumen [116]. The rearrangement of fluorescent fusion Ire1-GFP was observed following the treatment with non-thermal plasma. Such rearrangement is consistent with Ire1p activation, as a similar effect was also observed in control cells treated with DTT [100]. Moreover, in the cells treated with plasma, an increase in fraction of BiP that formed sedimentable complexes with unfolded proteins was observed [106]. To test whether plasma treatment indeed affects the traffic of proteins through the secretion pathway, the localization of Hsp30, a protein that is normally localized in the cytoplasmic membrane, was probed using a Hsp30-GFP fusion reporter. In plasma-treated cells, the fusion protein was found to mislocalize to the granules in the cell cytoplasm, which appears to be the result of disrupted transport of proteins from ER to the cytoplasmic membrane [100]. These observations together strongly indicate that the treatment of cells with non-thermal plasma leads to the stress in ER and affects the protein traffic in the secretory pathway.

Besides its effects on protein folding, the effects of plasma on translation were also investigated. Two types of aggregates that form in the cytosol of eukaryotic cells in the response to a stress that affects the translation - in yeast including heat shock, high concentration of ethanol, or treatment with sodium azide - are processing bodies (P-bodies) and stress granules (SG). While P-bodies contain translationally repressed mRNAs and a set of mRNA decay enzymes and are involved in mRNA degradation, SGs contain mRNA together with multiple components of the translation machinery and represent a storage form of mRNA that is not translated at the moment [117,118,119,120]. Formation of both types of aggregates, P-bodies and SGs, in plasma-treated cells was shown again by visualization of these aggregates by expression of GFP tagged proteins Dcp1-GFP, Dcp2-GFP (subunits of mRNA decapping complex) and Dhh1-GFP (mRNA decapping activator) as markers of P-bodies and Ngr1-GFP (RNA binding protein) and Pab1-GFP (poly(A) binding protein), markers of SGs [100]. Appearance of P-bodies implies that cell response to plasma includes a global repression of translation. Formation of SGs has been previously described to occur in yeast only under serious stresses such as glucose deprivation [121], high concentration of ethanol [122] and severe heat shock [123] or combination of mild stresses [124]. It is not clear which of the stresses produced by plasma induce the formation of SGs, but it is very likely that it is multiple different stresses that combine [100].

Together, these experiments show that besides the manifestation of typical hallmarks of stress, a response by several stress responsive pathways occurs in plasma-treated yeast (Figure 2). Deletion experiments described above together with these data show that the stress response pathways are in fact effective and their activation results in increased cell survival. This is further corroborated by the fact that pretreatment of cells with mild heat-shock, which activates most of these stress-response pathways, also improves yeast resistance to plasma [106]. Maintenance of cell survival by these stress response pathways, including increased plasma resistance of pretreated cell, likely results from transcriptional activation of many genes encoding for chaperones, enzymes involved in ROS detoxification and others. It is, thus, interesting that proteomic analysis of cells treated by plasma did not show an increased expression of stress-response genes [125]. As for now, there is no reasonable explanation for this contradiction. It needs to be addressed in the future to come.

## 8. Conclusions

Taken together, treatment with non-thermal plasma induces several types of stress in yeast cells. These stresses seem to be triggered by the presence of reactive particles, e.g., reactive oxygen and nitrogen species, that react with many cellular components, such as proteins, DNA and lipids. Plasma-induced damage of all of these components clearly contributes to cell killing. Although many targets of plasma have been described, there are not enough data yet to understand the contribution of individual defects to cell dying. Moreover, it is clear that while some of the observed effects of plasma are independent, e.g., direct chemical damage of multiple cellular components by reactive particles originating from plasma, others may be related. The relations between these effects, especially the casual relations, can be sometimes deduced from our knowledge of cell biology, but are generally hard to understand from the descriptive image of plasma-treated cells we have so far.

Other aspect of plasma treatment that is not yet understood is the contribution of individual reactive species to the damage of specific targets. Addressing this has been attempted by using scavengers specific for different reactive particles [47], as well as using yeast strains deficient in enzymes participating in specific detoxification pathways (e.g., superoxide by superoxide dismutases, see above). Using these or similar approaches will possibly give us a complex image of how cell components get damaged by individual chemical components of plasma and which response pathways are activated in response to what damage.

Several of the stress response pathways that have been shown to be activated in case of plasma treatment are known to respond to multiple types of stress. For example, the MAPK pathway involving Hog1p can be activated by oxidative stress, as well as by acidic stress [126]. As both of these types of stress are present in plasma treatment, it may be that either one of these two or both of them together play a relevant role in inducing the response to plasma. As of now, contributions of particular stresses have generally not been established. Even though at the first sight this may look as a nonproductive nitpicking, it may indeed have consequences in specific situations, for example when cells are treated with plasma in the buffered environment or in the presence of reactive particles scavenger.

We believe that yeast *S. cerevisiae* represents an excellent simple eukaryotic model, in which all of the outlined aspects of plasma effects on cells can be addressed and that our understanding of the processes in this organism will also be directly relevant to the effects of plasma on more complicated mammalian cells.

## Figures and Tables

**Figure 1 ijms-22-02247-f001:**
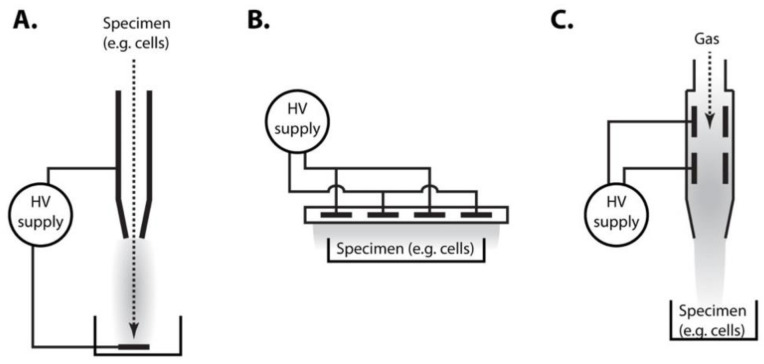
Typical experimental settings used for plasma treatment of cells. (**A**) Treated specimen is pumped through the hollow electrode and passes through electrical discharge, in which plasma (in all panels depicted in grey) is generated. (**B**) Plasma is generated by a discharge between electrodes separated by insulating dielectric barrier. Specimen is exposed to the plasma. (**C**) In plasma jet devices, plasma is generated between two electrodes inside a nozzle and streaming gas transmits plasma to the specimen. In all these settings, specimen may either contain treated cells (direct treatment) or certain type of medium (e.g., water), in which cells are subsequently incubated (indirect treatment).

**Figure 2 ijms-22-02247-f002:**
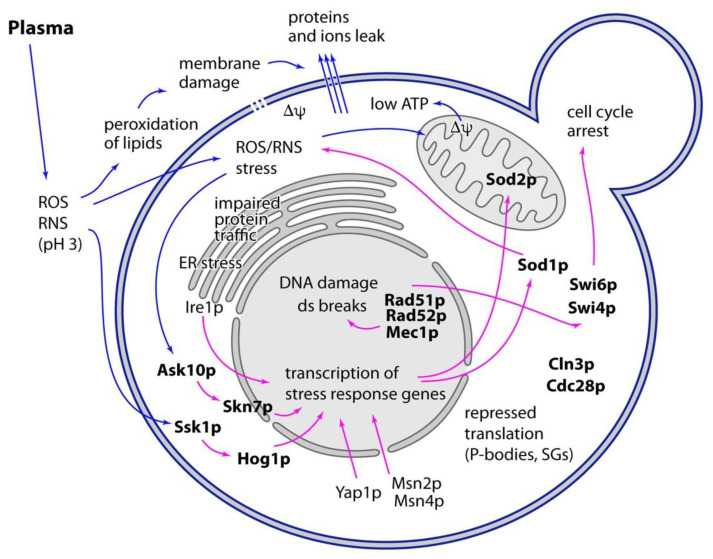
Summary of cold plasma effects on yeast cell. Treatment of cells with plasma induces multiple types of stress and damage (blue arrows). Cellular stress response pathways (purple arrows) act to protect cell from damage. Components of these pathways that are known to affect the cell survival (e.g., deletion of genes encoding for these components results in increased sensitivity of cells to plasma) are shown in bold.

## Data Availability

Not applicable.

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
