# Peer review of "Effects of Non-Thermal Plasma on Yeast Saccharomyces cerevisiae"

_ijms, 2021, doi:10.3390/ijms22052247_

Round 1

Reviewer 1 Report

In this review, authors summarized the research related to application of atmospheric pressure non-thermal plasma to yeast. Here are comments.

  1. Many small paragraphs are in text. It would be better to combine and reorganize those.
  2. Authors should check grammar errors and sentence readability.
  3. Authors should add more references in the text, particularly in molecular genetics description in yeast.
  4. Introduction

It is not clear why authors have reviewed on plasma application to particularly “yeast”.  It would be better to state an objective or focus of this review at the end of introduction.

  1. Section 2 Survival of yeast after plasma treatment
  • It is quite clear that cell survival depends on plasma dose. Is there any known information on quantitative relationship between survival and plasma dose? For example, standard dose for survival, medium survival and killing?
  • Authors should check grammar or readability in several sentences, for example, line 87, 92-93.
  1. Section 3 Plasma-induced changes in yeast cells
  • Section title seems too broad based on the content. Since the content is mostly focused on regulated yeast cell death, more specific title may be better.
  • References on apoptosis-like death in yeasts are few. Can authors be sure that this kind of death is real in yeasts?
  • Authors should check grammar or readability in several sentences, for example, line 128 “..are considered as the typical hallmarks..”, line 152-156.
  1. Line 280-281 and 299-301, add references.
  2. Line 334 and 336, GPF-labelle?, Yap1-GPF?
  3. Section 7 Plasma treatment induces multiple stress-response pathways

In description of molecular genetics on yeast pathways, many references are missing in the text. Please add references.

Author Response

Dear reviewer,

Thank you very much for reviewing our manuscript and for your valuable suggestions. In resubmitted version we are addressing your concerns. Please find a point-to-point answer below.

We believe that the manuscript in the present form is suitable for publication.

Yours sincerely

Peter Polcic

Reviewer 1

In this review, authors summarized the research related to application of atmospheric pressure non-thermal plasma to yeast. Here are comments.

  1. Many small paragraphs are in text. It would be better to combine and reorganize those.

    We did rephrase several paragraphs, especially the shorts ones. We agree with the reviewer that this makes the manuscript more readable.

  1. Authors should check grammar errors and sentence readability.

We reviewed the language of the manuscript and we believe that the grammar and readability are improved in present form. Most of the edits in the text of the manuscript (except of those specifically listed in the answer to other points) were done for the purpose of addressing this issue.

  1. Authors should add more references in the text, particularly in molecular genetics description in yeast.

References have been added. Some of the references are repeating but also 22 new references were included.

  1. Introduction

It is not clear why authors have reviewed on plasma application to particularly “yeast”.  It would be better to state an objective or focus of this review at the end of introduction.

Yeast S. cerevisiae are a traditional model organism that has many advantages over the more complicated mammalian cells. General knowledge of most of the processes occurring in eukaryotic cells comes from exploiting yeast as a model (we name several important examples in Introduction). We and many other authors believe that yeast are also a suitable model for studying interactions of plasma with eukaryotic cells. This is also mentioned in the second paragraph of Introduction. We agree with the reviewer that this paragraph needed a better conclusion. We therefore included concluding sentences.  

  1. Section 2 Survival of yeast after plasma treatment
  • It is quite clear that cell survival depends on plasma dose. Is there any known information on quantitative relationship between survival and plasma dose? For example, standard dose for survival, medium survival and killing?

The term "plasma dose" has been a subject of many discussions in the community of plasma medicine and other plasma biomedical application. It issues from the apparent need to quantify the "dose" character of the plasma treatments of biological cells, tissues, liquids, etc. Quite naturally, a lower "plasma dose" is expected to induce weaker effects than a higher "dose", or more interestingly, various biological targets may have a "threshold dose" from which the plasma treatment results in a certain effect (e.g. bacterial inactivation or apoptosis induction in cancer cells) and below which this effect is not observed. Or, there can be even something similar to the lethal "dose" from which the plasma treatment results in a severe damage of cells/tissues (e.g. necrosis or tissue burn).

The difficulties in clearly defining the plasma dose are: 1) cold plasma is neither purely physical phenomenon, such as radiation or electric field, nor purely chemical mixture of radicals and other RONS;

2) there are  many different cold plasma sources that produce plasmas of different properties; with different intensities of UV radiation and E-fields and different compositions, concentrations and ratios of their chemical components.

If plasma was regarded just like a physical factor, such as radiation, its intensity, wavelength and treatment time would determine the dose. Similarly, if it was just (pulsed) E-field, its intensity, pulse duration and repetitive frequency, and treatment time would define a dose.

If the plasma effect were just chemical, then similar to the dose of chemical agents, it would be determined by concentration x time (e.g. Wright et al., Quantification of the Ozone Dose Delivered into a Liquid by Indirect Plasma Treatments: Method and Calibration of the Pittsburgh Green Fluorescence Probe, Plasma Chemistry and Plasma Processing, 2018). This is applicable e.g. for plasma activated water and liquids applied to cells post plasma treatment if exact concentrations of reactive species are measured therein. It is not straightforward either, since many reactive species have very short lifetimes and the species interact inside the liquid.

Since there are always multiple plasma agents in direct plasma treatments, the definition of a single plasma dose is not clear. Several approaches have been suggested how to express the relative "dose": 

1) plasma treatment time only (e.g. Barekzi and Laroussi: Dose-dependent killing of leukemia cells by low-temperature plasma, 2012 J. Phys. D: Appl. Phys. 45 422002)

2) plasma electrical power (which relates somehow to the intensity of the E-field, radiation and concentration of RONS, etc. but these strongly depend on the type of plasma discharge and its gas/liquid environment)

3) power x time (i.e. delivered energy)

None of these, however, does clearly and uniformly express the physico-chemical effects of the plasmas and the most meaningful definition of the “plasma dose” is still under investigations [Wende et al., Chemistry and biochemistry of cold physical plasma derived reactive species in liquids, Biol. Chem. 2019; 400(1): 19–38) 

In the revised manuscript, we included a note and one reference so that the relevant sentence now reads: “Sensitivity of eukaryotic cells, including yeast, to plasma treatment is generally lower as compared with bacteria, but significant fraction of eukaryotic cells can be killed by cold plasma in a dose-dependent manner (with plasma “dose” usually expressed as the plasma treatment time) [18-20].”

  • Authors should check grammar or readability in several sentences, for example, line 87, 92-93.

Most of the paragraph (lines 87 to 93 of original manuscript) was rephrased. We believe, the corrections improved the grammar and readability.

The paragraph now reads: “As one would expect, significantly more yeast cells are killed by direct plasma treatment than when cells are incubated in plasma-activated water [25]. These results indicate that in direct treatment regime, a significant portion of cells dies immediately due to the exposure to the factors present in the electric discharge (e.g. electric field, short-living reactive particles, UV irradiation). Additionally, a portion of cells that survive the discharge dies due to the presence of reactive particles generated in the discharge and surviving in the plasma-activated water for significant periods of time after the discharge. From a comparison of survival rates of directly and indirectly treated cells, it appears that these two components of cell killing effects act together, likely in synergy.”

  1. Section 3 Plasma-induced changes in yeast cells
  • Section title seems too broad based on the content. Since the content is mostly focused on regulated yeast cell death, more specific title may be better.

The title has been changed to “Plasma-induced cell death in yeast”

  • References on apoptosis-like death in yeasts are few. Can authors be sure that this kind of death is real in yeasts?

First papers describing apoptosis like cell death appeared in late nineties of the 20th century. Since then it has been well established that yeast do undergo regulated forms of cell death under various situations. We included references to several influential review articles (references 31-35).  

  • Authors should check grammar or readability in several sentences, for example, line 128 “..are considered as the typical hallmarks..”, line 152-156.

As in the other sections of the text, we believe we improved the grammar and readability, including the suggested change.

The latter sentence now reads: “Because in apoptosis, or in other forms of programmed cell death, cells are dying due to the execution of intrinsic cell killing program rather than directly due to the damage, to prove that apoptosis participates in cell killing by plasma, it has to be shown that blocking of participating cell death pathway would result in an increased cell survival after treatment with plasma.”

  1. Line 280-281 and 299-301, add references.

Reference has been added. It is reference no. 78. 

  1. Line 334 and 336, GPF-labelle?, Yap1-GPF?

   These typos have been corrected to ‘GFP-labelled’ and ‘Yap1-GFP’.

  1. Section 7 Plasma treatment induces multiple stress-response pathways

In description of molecular genetics on yeast pathways, many references are missing in the text. Please add references.

    We added references to the text. As already mentioned, some of them are newly introduced and some are references were already included elsewhere in text.

Reviewer 2 Report

Comments to Authors: In this review, the authors clearly describe the critical effects, signal transductions, pathways, and reactions of various kinds of plasma treatments to the cells, especially yeast, and the relationship of the responsive molecules with the plasma treatments. This review will provide enough information to the readers to think about their future studies. The contents of this review are valuable, and suit for publication in International Journal of Molecular Sciences after minor revisions listed below.   1. At page 3, on line 131: The reviewer thinks that “phosphatidyl serine” may be incorrect, it should be “phosphatidylserine”. The authors should confirm that. 2. At page 3, on line 133 “DAPI”, and on line 134 “TMRM”, and at page 4, on line 177 “JC-1”, and at page 7, on line 355 “STRE”. For readers, the authors should explain these abbreviations, for example “malonedialdehyde (MDA)” on line 163 at page 4, or put them in the lists of “Abbreviations” at page 10. 3. At page 7, on line 334, “GPF-labelled” should be “GFP-labelled”? On line 336, “Yap1-GPF” should be “Yap1-GFP”?

Author Response

Dear reviewer,

Thank you very much for reviewing our manuscript. Please find the point-by-point response to your concerns below.

Reviewer 2

Comments to Authors: In this review, the authors clearly describe the critical effects, signal transductions, pathways, and reactions of various kinds of plasma treatments to the cells, especially yeast, and the relationship of the responsive molecules with the plasma treatments. This review will provide enough information to the readers to think about their future studies. The contents of this review are valuable, and suit for publication in International Journal of Molecular Sciences after minor revisions listed below.   

  1. At page 3, on line 131: The reviewer thinks that “phosphatidyl serine” may be incorrect, it should be “phosphatidylserine”. The authors should confirm that.

 “phosphatidyl serine” has been corrected to “phosphatidylserine”

  1. At page 3, on line 133 “DAPI”, and on line 134 “TMRM”, and at page 4, on line 177 “JC-1”, and at page 7, on line 355 “STRE”. For readers, the authors should explain these abbreviations, for example “malonedialdehyde (MDA)” on line 163 at page 4, or put them in the lists of “Abbreviations” at page 10. 

For DAPI and TMRM, the chemical names have been inserted. JC-1 is a potential-sensitive probe. To our knowledge, the abbreviation in this cease does not correspond to a chemical name. All three are fluorescent probes that are mostly known under abbreviated names. 

DAPI, TMRM and STRE have been also added to the list of abbreviations, as you suggested. 

  1. At page 7, on line 334, “GPF-labelled” should be “GFP-labelled”? On line 336, “Yap1-GPF” should be “Yap1-GFP”?

This was a mistyping. It has been corrected.

Besides these corrections we have also edited a text of the manuscript for better grammar and readability without affecting the meaning. We believe that the manuscript in the present form will be suitable for publication.

Yours sincerely,

Peter Polcic

Round 2

Reviewer 1 Report

Authors answered all questions and modified the text addressing reviewer’s comments. I think the manuscript is acceptable for publication.